# Enthalpy-entropy compensation of atomic diffusion originates from softening of low frequency phonons

Simon Gelin [1,2,3 ✉], Alexandre Champagne-Ruel [1] & Normand Mousseau [1 ✉]

Experimental data accumulated over more than 120 years show not only that diffusion coefficients of impurities ordinarily obey the Arrhenius law in crystalline solids, but also that diffusion pre-exponential factors measured in a same solid increase exponentially with activation energies. This so-called compensation effect has been argued to result from a universal positive linear relationship between entropic contributions and energy barriers to diffusion. However, no physical model of entropy has ever been successfully tested against experimental compensation data. Here, we solve this decades-old problem by demonstrating that atomistically computed harmonic vibrational entropic contributions account for most of compensation effects in silicon and aluminum. We then show that, on average, variations of atomic interactions along diffusion reaction paths simultaneously soften low frequency phonons and stiffen high frequency ones; because relative frequency variations are larger in the lower region of the spectrum, softening generally prevails over stiffening and entropy ubiquitously increases with energy.

[1] Département de physique and Regroupement québécois sur les matériaux de pointe, Université de Montréal, C.P. 6128, Succursale Centre-Ville, Montréal, Québec H3C3J7, Canada. [2] Département de mathématiques et de génie industriel, École Polytechnique de Montréal, Montréal, Québec H3C3A7, Canada. [3] Institut Lumière Matière, UMR5306 Université Lyon 1-CNRS, Université de Lyon, F-69622 Villeurbanne Cedex, France. ✉email: simon.gelin@univ-lyon1.fr; normand.mousseau@umontreal.ca

Atomic diffusion in solids proceeds via sequences of thermally activated atomic jumps, which, in the presence of chemical inhomogeneities, induce a flux of atoms that can be described at the mesoscopic level by the diffusion coefficient $D$[1,2]. In crystals, where sequences of diffusive events are usually dominated by a single-jump mechanism, $D$ is proportional to the product of the average rate at which jumps are activated and the concentration of defects possibly mediating jumps. Drawing on reaction rate[3] or dynamical[4] theories to compute the average jump rate, it has become customary to interpret the temperature dependence of $D$ using the semi-empirical relation,
$$D(T) = g\, f\, \nu \exp((\Delta S_{\mathrm{m}} + \Delta S_{\mathrm{f}})/k_B) \exp(-(\Delta E_{\mathrm{m}} + \Delta E_{\mathrm{f}})/k_B T),$$
where $g$ and $f$ are geometrical and correlation factors, respectively, $\nu$ an attempt frequency, $\Delta E_{\mathrm{m}}$ and $\Delta E_{\mathrm{f}}$ are the enthalpy barriers (energy barriers at zero pressure) to atomic migration and defects formation, respectively, and $\Delta S_{\mathrm{m}}$ and $\Delta S_{\mathrm{f}}$ are the corresponding entropic contributions[2]. Although resting on thermodynamical principles and being compatible with the Arrhenius law, this relation remains phenomenological if not complemented by a mechanistic framework within which to evaluate and interpret the different terms it is composed of. Establishing such a framework is particularly difficult for entropic contributions, as they may arise from diverse physical phenomena at the atomic or electronic scales[5–8], without any general rule to assess which dominates.

This issue is best exemplified by the limited understanding of the ubiquitous compensation effect, also referred to as Meyer-Neldel rule[9,10], according to which diffusion pre-exponential factors $D_0$ of different impurities diffusing in a same solid increase exponentially with their activation energy $\Delta E$: $D_0 = D_{00} \exp(\gamma_{\mathrm{c}} \Delta E)$, with $D_{00}$ and $\gamma_{\mathrm{c}}$, the compensation pre-exponential factor and compensation factor, respectively. This so-called "law of compensation" has been reported in a wide variety of solids, including metals[10], minerals[11], semiconductors[12], and ionic crystals[13]; we illustrate it in Fig. 1a for self- and impurity diffusion in silicon and aluminum. In light of the semi-empirical relation, this law suggests the existence of a fundamental positive linear relationship, called enthalpy–entropy compensation, between entropic contributions and energy barriers to diffusion: $\Delta S_{\mathrm{m,f}} = k_B \gamma_{\mathrm{c}} \Delta E_{\mathrm{m,f}}$, with $\gamma_{\mathrm{c}} > 0$. However, it remains controversial whether enthalpy–entropy compensation originates from an underlying general physical principle[9], or simply results from trifling experimental errors[14,15].

The two most popular explanations for the compensation of energy barriers by entropy are based on phenomenological models of migration entropy: Zener's model, drawing on reaction rate theory, ascribes compensation to a loosening of crystalline lattices' elastic moduli at transition states (TSs)[16], while the multiexcitation entropy model explains the increase of entropic contributions as resulting from the increasing number of ways phonons can assemble to overcome higher energy barriers[9]. Unfortunately, because they resort to qualitative descriptions of atomic diffusion, both models introduce arbitrary parameters that make them untestable, so that it remains unknown whether any of them identifies the correct physical origin of compensation.

To solve this issue, we systematically compare atomistic simulations of harmonic diffusion pre-exponential factors with previously published experimental compensation data, in silicon and aluminium single crystals. This comparison shows that harmonic vibrational entropy accounts for ≳70% of compensation effects in both materials, demonstrating that the physical origin of compensation is mostly contained in this level of mechanistic description. Drawing on this discovery, we perform a detailed statistical analysis of entropy variations as a function of potential energy along elementary-activated events in four different amorphous solids. The disorder of amorphous solid structures gives access to hundreds of thousands of events, with continuously dispersed activation energies, and allows us to unveil the generic mechanism at the origin of the compensation law. As a solid is deformed along a diffusion reaction path, some of the atomic bonds are stretched (and eventually broken), while others are compressed (to provide the moving atoms with space for their motion). On average, these two mechanisms simultaneously soften low-frequency vibrational modes of the solid (increase entropy) and stiffen high-frequency ones (decrease entropy). This broadening of the spectrum, which intensifies as the activation energy increases, generally leads to compensation because entropy changes are controlled by relative variations of modes' frequencies, and these are larger at low frequencies. However, we exhibit counterexamples where stiffening is so

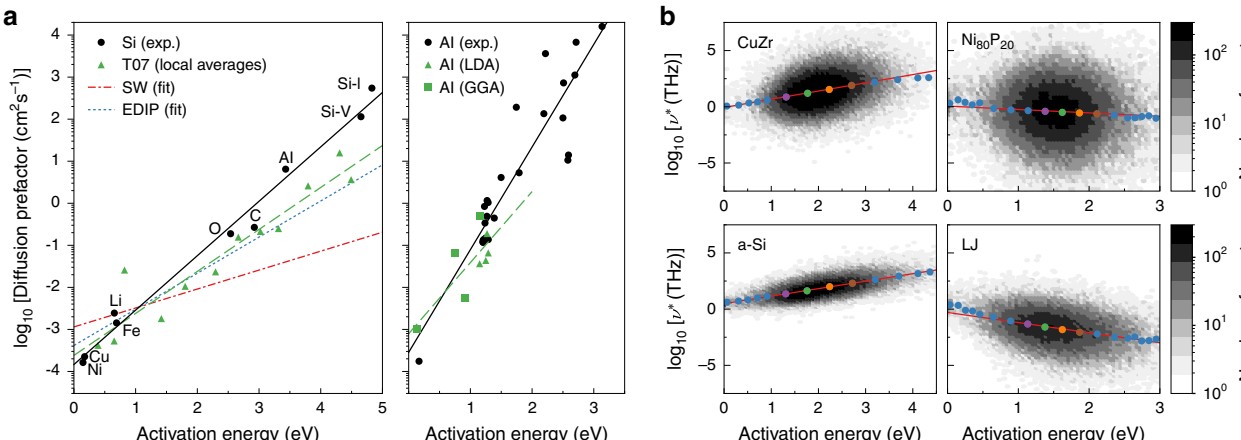

**Fig. 1 Compensation in silicon, aluminium, and four amorphous solids. a** Compensation effect for diffusion in crystalline silicon (left) and aluminium (right) (experimental and numerical data are all reported in Supplementary Information, the computation of local averages and the fitting procedure are explained in "Methods"). The compensation factor is equal to 3.0 eV$^{-1}$ in silicon, and 5.6 eV$^{-1}$ in aluminium. The Stillinger–Weber, EDIP and modified Tersoff silicon models give compensation factors of 1.0, 2.0, and 2.3 eV$^{-1}$, respectively. Density functional theory data in aluminium—obtained with the local density approximation (LDA) or the generalized gradient approximation (GGA)—also obey compensation, with a compensation factor of 3.8 eV$^{-1}$. **b** Compensation effect for harmonic activation rates in four amorphous solids: a-Si, CuZr, Ni$_{80}$P$_{20}$, and LJ; compensation factors are extracted from linear fits to all data and equal 1.53, 1.68, −0.70, and −2.07 eV$^{-1}$, respectively. Dots represent local averages of activation energies and $\log_{10}(\nu^\star)$ values over non-uniform energy bins; their color enables to distinguish transition states investigated in Fig. 2 based on the energy bin they belong to.

intense that it dominates softening, and anti-compensation is observed. Our analysis allows us to resolve the puzzling fact that the nature of local impurities has no significant statistical impact on compensation since properties of low frequency, extended vibrations—which govern entropy variations—are controlled mainly by the host solid; this is why a single-compensation law is usually observed in a given solid.

## Results

**Numerical vs. experimental data in crystals**. Our numerical study starts with the observation that all Arrhenius parameters of diffusion processes reported in Fig. 1a fall on a single straight line, extending over more than three electronvolts in activation energies and seven orders of magnitude in pre-exponential factors. This remains true whether diffusion proceeds via the direct interstitial mechanism (Ni, Cu, Li, Fe, O in Si, and H in Al) or via defect-mediated mechanisms. This observation entails that compensation parameters, $D_{00}$ and $\gamma_c$, do not depend on the specific nature of diffusing impurities, but rather on physical properties of the host solid lattice. It also implies that variations of migration and formation entropies share the same origin. Therefore, the experimental compensation effect may be fully captured within numerical simulations of migration barriers to self-diffusion in pure host solids. We do so here by sampling TSs (saddle points) surrounding the ground state (GS) of a computational model of crystalline silicon-containing defects, using the activation relaxation technique nouveau (ARTn) algorithm[17,18] ("Methods"). The creation of defects (vacancies and self-interstitials) gives us access to a richer set of activated events. We then compute, for each TS, the pre-exponential factor $\nu^\star = \nu \exp(\Delta S_m/k_B)$ of the average rate at which these TSs are crossed within the harmonic transition state theory (hTST)[19]: $\nu^\star = \prod_i \nu_i / \prod_i \nu_i^\ddagger$, where $\nu_i$ and $\nu_i^\ddagger$ denote positive vibrational frequencies at GS and TS, respectively. Similarly to the expression of the harmonic vibrational entropy of formation[20], the hTST prefactor derives from an approximation of the solid's canonical partition functions, here at the GS and at the TS, based on a second-order expansion of the mass-rescaled potential energy landscape (PEL). Therefore, although the hTST formulation does not allow for a unique definition of the semi-empirical attempt frequency and migration entropy, it suggests by consistency with the formation entropy expression, a decomposition where the attempt frequency captures an average vibrational property of the solid's GS, while migration entropic contributions are controlled by variations of local curvatures of the PEL when moving from the GS to the TS.

Finally, we test our results against experimental data by assigning a diffusion coefficient to each sampled event, with $D_0$ the product of the hTST-crossing rate and the typical squared jump length of the most displaced atom, and $\Delta E$ the potential energy difference between the TS and the GS. To ensure the generality of our results, we repeat the procedure with three empirical potentials developed for silicon. Numerical results follow the compensation law for all studied empirical potentials (Fig. 1a, left, discontinuous lines). In particular, the recent parametrization based on Tersoff's model ("Methods") reproduces 77% of experimental compensation effects (green triangles in Fig. 1a, left). To provide an independent assessment of the contribution of vibrational entropy variations to compensation, we compare experimental diffusion data in aluminum with previously published first-principles computations relying on the harmonic approximation ("Methods"). Despite the small range of activation energies probed numerically, most of the compensation effect—~70%–is accounted for by variations of vibrational entropy (Fig. 1a, right). In addition to numerical errors (finite-

size effects in first-principles calculations or missing features of atomic interactions in empirical models), it is likely that additional increases of entropy may require the inclusion of electronic contributions[7,8] and, possibly, of the anharmonicities of the solids' PEL.

**Generic broadening of VDOS at transition states**. Even considering these possible additional contributions, our results demonstrate clearly that enthalpy–entropy compensation originates largely from harmonic vibrational contributions to entropy. However, they build on a restricted set of activated events and systems and, therefore, are not sufficient to disentangle the general features of vibrational changes at the origin of the compensation effect from event-specific properties leading to the dispersion of the data around the compensation line. To lift this limitation, we turn to activated events around inherent states (ISs) of disordered materials, which give access to a potentially astronomical number of different configurations and activation barriers. We focus on four prototypical amorphous solids: silicon (a-Si), CuZr, $Ni_{80}P_{20}$, and a 2D binary Lennard–Jones (LJ) mixture ("Methods"). These cover a wide range of bonding and local environments, from elemental covalent amorphous materials (a-Si) to metal–metal (CuZr), metal–metalloid ($Ni_{80}P_{20}$), and hard 2D glasses (LJ). As shown in Fig. 1b, activation energies of events sampled in these solids with ARTn spread continuously over several electronvolts, with hTST activation rate prefactors varying exponentially with energies. While there is a large dispersion on an event per-event basis, we find clear trends when averaging over these large data sets: prefactors increase in CuZr and a-Si, according to the compensation law, and decrease in $Ni_{80}P_{20}$ and LJ, showing an anti-compensation that has already been observed in a non-physical LJ glass[21]. These diverse responses provide useful data for developing a quantitative explanation for the correlation between prefactors and energy barriers.

Since, as we have demonstrated, compensation largely originates from variations of VDOS, we investigate frequency changes as the system moves from an IS to a neighboring TS by averaging VDOS over data sets of >100,000 events sampled in 50 independent realizations per solid type. This procedure gives unprecedented access to the core signal associated with frequency transformations. In Fig. 2a, we represent normalized VDOS at ISs and TSs. Given that only a few tens of atoms are significantly displaced at TSs, and studied systems contain ~4000 atoms, individual differences are hardly distinguishable. We thus report in Fig. 2b the cumulative differences of (non-normalized) VDOS between TSs and ISs, averaged over TSs in different activation energy intervals ("Methods"). A systematic trend, common to all solids, clearly emerges: cumulative VDOS differences increase in the lower region of the spectrum, indicating a softening of low-frequency phonons at TSs with respect to ISs; a similar trend is observed at the high end of the spectrum (where signals converge to minus one due to the presence of a negative-frequency mode at TSs), caused this time by a hardening of some vibrational modes at TSs. These co-occurring shifts at the extremities of the spectrum create a depletion of modes in the center, responsible for a rapid decrease of cumulative VDOS differences in this region. The physical origin of the broadening of the spectrum is clear: moving from a local point of stability to a transition state both shortens and elongates interatomic distances and angles around the geometrical center of the activated event, which in turn perturbs the distribution of dynamical matrix components with both softer and stiffer atomic interactions. On average, softer interactions amplify the proportion of low-frequency, extended vibrations while stiffer ones impact high-frequency, localized excitations. We further confirm this scenario by showing that

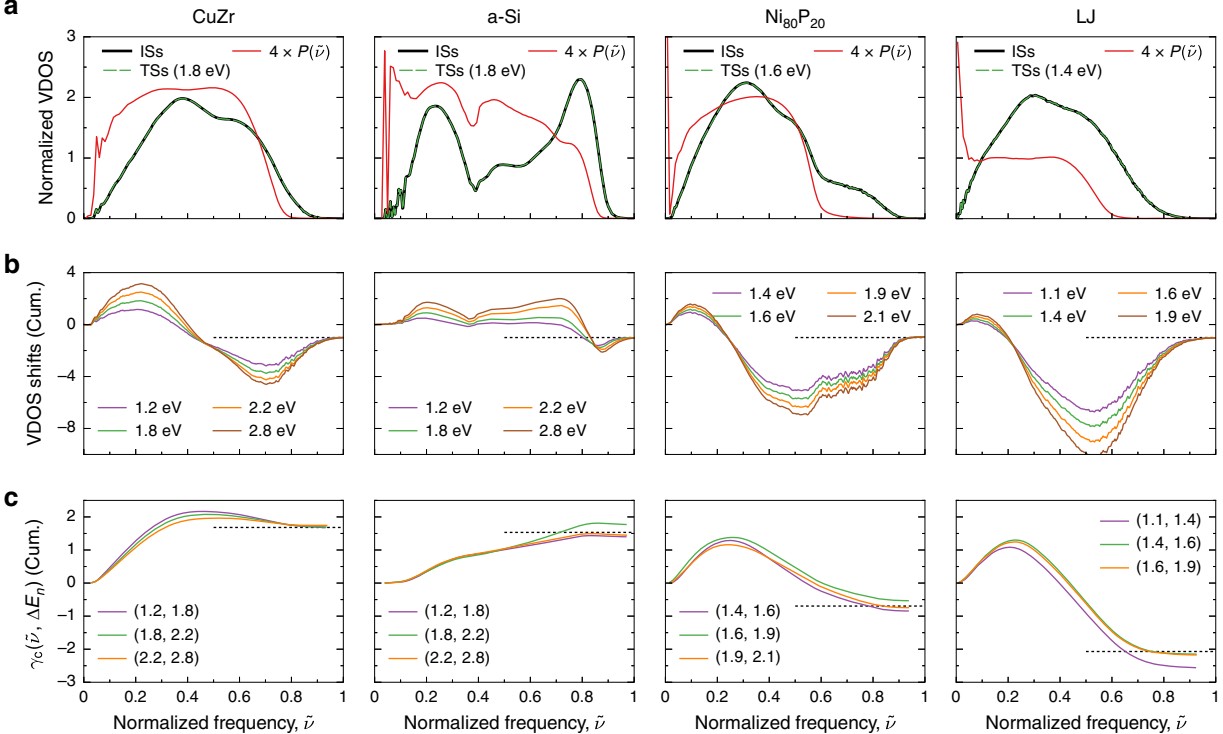

**Fig. 2 Vibrational origin of compensation and anti-compensation. a** Normalized VDOS averaged over all ISs (black lines) and most probable TSs (green dashed lines), as a function of normalized frequencies $\tilde{\nu} = \nu/\nu_0$, with $\nu_0$ equal to 9, 20, 15, and 7.5 THz for CuZr, a-Si, Ni80P20, and LJ, respectively. Red lines represent mode participation ratios ("Methods"), averaged over frequencies and ISs, and multiplied by four so that their values can be read on the y axis used for normalized VDOS. **b** Cumulative VDOS shifts between TSs— whose activation energy lies in the energy bin centered at $\Delta E_n$—and ISs. They converge to −1 (black dotted lines). Legends indicate approximate values of $\Delta E_n$, and line colors correspond to colors of dots in Fig. 1b. **c** Cumulative contributions of vibrational modes to local compensation factors $\gamma_c(\tilde{\nu}, \Delta E_n)$ (slopes of the line connecting two consecutive dots in Fig. 1b; these dots lie around the energies $(\Delta E_n, \Delta E_{n+1})$ given in legends). They converge around the global compensation factors $\gamma_c$ (black dotted lines).

cumulative VDOS differences systematically reach their minimum value at a frequency close to the last inflection point of the participation ratio curves, from where it drops to zero and thus modes localize. The broadening described here is perfectly consistent with VDOS deformations induced by isotopic impurities in model crystalline lattices[22] as well as perturbations of spring constants in disordered solids models[23].

**How does compensation emerge?** To understand how compensation emerges, we represent in Fig. 2c the logarithm of the cumulative product of average-frequency ratios $\nu_{i,n}^{\ddagger}/\nu_{i,n+1}^{\ddagger}$—with $i$ the indexes of modes ordered by ascending frequency, and $n$ the indexes of the energy bin over which frequencies are averaged—, divided by the difference of activation energies of TSs lying in bins centered at $\Delta E_{n+1}$ and $\Delta E_n$ ("Methods"). This quantity, that we call the cumulative local compensation factor, $\gamma_c(\nu, \Delta E_n)$, converges toward the slope of the line connecting consecutive local averages of energy-prefactor data represented by circles in Fig. 1b. Cumulative compensation factors start by increasing, because the piling up of low-frequency modes at the bottom of the spectrum systematically increases with activation energy in all solids (Fig. 2b). Inversely, they end up decreasing at the top of the frequency spectrum, where high-frequency modes of high-energy TSs spread over higher values than high-frequency modes of low-energy TSs. Therefore, (anti-)compensation in a-Si (Ni$_{80}$P$_{20}$, LJ) originates in the asymmetric broadening of VDOS at TSs, which causes positive (negative) contributions to the compensation factor to operate over most of the spectrum. In CuZr, the competition between softening and stiffening opposite contributions is subtler; cumulative compensation factors reach their maximum

before the center of the spectrum but then do not decrease fast enough to counteract the initial increase. This is because a fixed-frequency shift $\nu_{i,n}^{\ddagger} - \nu_{i,n+1}^{\ddagger}$ has a lower impact on the frequency ratio $\nu_{i,n}^{\ddagger}/\nu_{i,n+1}^{\ddagger}$ controlling the cumulative compensation factor as the frequency $\nu_{i,n}^{\ddagger}$ increases. This dominance of delocalized modes explains the ubiquity of the compensation effect: as energy barriers increase, the broadening of the spectrum intensifies, and softening, which increases entropy, becomes more and more important than stiffening. It also explains the weak impact of the nature of the diffusing impurity on the compensation law. Indeed, while the change of localized vibrational modes frequencies upon activation may be significantly affected by interactions between host solid atoms and impurities, that of delocalized modes frequencies is controlled mainly by the properties of the host solid.

## Discussion

From a physical point of view, the difference between various systems emerges both from variations in solids microstructure and the details of atomic interactions. On average, directional covalent bonds in amorphous silicon limit local coordination number and, thereby, result in a very open structure where activation typically proceeds through the stretching of bonds, which softens vibrational modes. Since higher barriers are generally associated with more bonds being stretched further, additional softening accompanies them, explaining the origin of the compensation effect in this solid. In contrast to this behavior, the hard 2D LJ solid is a dense structure with rigid interactions. As there is little free volume around atoms, activation tends to involve collisions that, contrarily to silicon, will stiffen

interactions and shift vibrational frequencies to higher values; here, higher energy barriers are largely associated with more atoms moving through even tighter environments, hence generating an anti-compensation correlation. For the two other systems, the balance between softening and stiffening of the vibrational spectrum cannot clearly be deduced from a general overview of the nature of interactions and the description of their variations during activation; the diversity of local atomic environments gives rise to considerable dispersion of harmonic prefactors around the compensation law—spanning more than five orders of magnitude in CuZr and ten in $Ni_{80}P_{20}$ (Fig. 1b)—, so that only a quantitative evaluation of harmonic prefactors over all microscopic events, such as the one performed here, can confirm the sign and magnitude of the average enthalpy–entropy correlation.

To conclude, our work establishes that the compensation effect for diffusion in crystals is not a mere experimental artifact; it is reproduced quantitatively within atomistic simulations, and originates mainly from the generic broadening of solids vibrational spectra caused by the dispersion of atomic interactions at perturbed states. It confirms Zener's general idea that lattice loosening can cause enthalpy–entropy compensation, and goes well beyond this proposition by unraveling in detail how loosening emerges from the unbalanced competition between mode softening at the low end and stiffening at the high end of the vibrational spectrum.

This analysis allows us to predict that anti-compensation may only be observed in solids whose atoms are densely packed and held together through stiff interactions, and whose vibrational spectrum contains an unusually large proportion of high-frequency localized modes. We exhibit such an example with amorphous $Ni_{80}P_{20}$. In addition, the fundamental nature of the softening mechanism revealed here suggests that it extends beyond atomic diffusion and explains the ubiquity of enthalpy–entropy compensation relations for processes controlled by a single microscopic-free energy barrier, observed across disciplines, including materials science[9,24], chemistry[25,26], and biology[27].

## Methods

**Experimental diffusion parameters in crystalline silicon and aluminum.** The compensation effect is commonly observed for different impurities diffusing in a same solid. It also holds for a given impurity diffusing in similar solids, but it remains unclear how to establish whether two solids are similar or not[28]. We thus focus here on the former situation. In addition, we restrict ourselves to single crystals, and more specifically to diffusion processes that are governed by a single-jump mechanism, so that compensation effects necessarily emerge from the relation between energetic and entropic properties of both the jump and the concentration of defects mediating diffusion, if any. In these conditions, we collect, in silicon single crystals, the data for species diffusing via the direct interstitial mechanism, the kick-out mechanism, the interstitialcy mechanism, and the vacancy mechanism. For each species, we assemble diffusion coefficients from the literature and fit them with the Arrhenius law to extract diffusion parameters. Collected diffusion coefficients and Arrhenius fits are reported in Supplementary Fig. 1. In aluminum single crystals, we gather diffusion parameters for species diffusing via the direct interstitial mechanism, and the vacancy mechanism. We report, in Supplementary Information, the exhaustive list of references we used to collect the data in silicon and aluminum, and provide in Supplementary Table 1 all diffusion parameters displayed in Fig. 1a.

**Ground states of crystalline silicon with defects.** The analysis of experimental compensation data in silicon (see the main text) shows that kinetic and thermodynamic compensation effects are quantitatively equivalent and do not depend on the nature of diffusing impurities. We therefore study compensation in crystalline silicon configurations containing different vacancy or self-interstitial structural defects—to access a richer set of activated events[29]—but no chemical impurity. We simulate these pure silicon systems with three of the most widely used empirical descriptions of Si atoms interactions: the three-body Stillinger and Weber model[30] (SW), the environment-dependent interatomic potential[31] (EDIP), and a recently modified version of the bond order Tersoff potential[32] (T07). To create vacancy and interstitial defects, we start by removing one atom or two neighboring atoms, or

inserting one atom or two atoms at neighboring positions, in diamond cubic crystals containing 4096 atoms. We then relax the resulting defective configuration during 25 ns, at 800 K and at a fixed density corresponding to the crystal equilibrium density at zero pressure and 800 K ($\rho_{SW} = 2.309$ g cm$^{-3}$, $\rho_{EDIP} = 2.322$ g cm$^{-3}$, and $\rho_{T07} = 2.296$ g cm$^{-3}$), while generating local points of stability by energy minimization every 0.25 ns, with the criterion that any component of the force field is lower than $f_{max}^{IS} = 3 \times 10^{-8}$ eV Å$^{-1}$. We estimate the ground-state (GS) configurations as those of lowest potential energy among thereby probed local minima.

**Inherent states of amorphous solids.** We study amorphous silicon with the modified Tersoff potential used for crystalline silicon, CuZr and $Ni_{80}P_{20}$ with embedded atom method (EAM) potentials[33,34] distributed on the eampotentials website (version 10/5/2011 for CuZr and version 10/18/2011 for $Ni_{80}P_{20}$). The LJ mixture is composed of 55% of small particles, of radius $r_S = 0.75$ Å, and 45% of large particles, of radius $r_L = 1.25$ Å, which interact through the following potential: $V_{ij}(r_{ij})/\epsilon = 1/x_{ij}^{12} - 2/x_{ij}^{6} + \alpha x_{ij}^{4} + \beta x_{ij}^{2} + \gamma$, where $\epsilon$ is the energy scale, $x_{ij} = r_{ij}/(r_i + r_j)$, and $\alpha$, $\beta$, and $\gamma$ are constant parameters set to ensure first and second derivatives of $V_{ij}$ are continuous at the cutoff $x_{ij}^c = 2$ ($\alpha = 747/65$, 536, $\beta = -117/1$, 024, and $\gamma = 313/1$, 024). Molar masses of both small and large particles are equal to 50 g mol$^{-1}$. To generate inherent states of amorphous solids, we equilibrate a liquid melt at $T_{eq}$ that we then quench down to $T_{relax}$ at a constant quench rate $q$ (values of the different parameters are given in Supplementary Table 2). Finally, we let the obtained configuration relax at $T_{relax}$ for a time $t_{relax}$. These steps are performed at zero pressure, except in the case of LJ, whose density is fixed at $\rho_N = N/L^2 = 0.2832$ Å$^{-2}$. Finally, inherent states are obtained by energy minimization at constant volume, with the criterion that any component of the force field is lower than $f_{max}^{IS}$. Contrarily to the other systems, the LJ mixture reaches equilibrium at $T_{relax}$ (we let this system relax for more than 3000 ns at ~980 K, a temperature at which the relaxation time of shear stress was estimated to be around 0.5 ns in an almost identical model[35]). For each solid type, 50 independent inherent states are generated. Crystalline GSs and amorphous ISs were prepared using LAMMPS[36].

**Sampling of activated events.** Starting from crystalline GSs or amorphous ISs, we sample reaction paths connecting two neighboring local minima of the potential energy landscape (PEL) using the open-ended saddle point search method, ARTn[17,18]. In amorphous solids, we only probe events centered around a limited set of atoms. To do that, we randomly pick atoms lying in a cube of volume $N_{ac}/\rho_N$, with $N_{ac}$ given in Supplementary Table 2, and $\rho_N$ the number density, then we randomly displace all atoms lying within a distance $R_a$ from the picked (center) atom, away from the initial minimum until a direction of negative curvature of the PEL is found. We decompose a reaction path in three configurations—the initial minimum (the GS or IS), the saddle point, and the final minimum—, and impose that the maximum absolute value of any force component is lower than $f_{max}^{IS}$ at initial and final minima and lower than $f_{max}^{\ddagger}$ at the saddle point (see Supplementary Table 2 for the values of these different parameters). To ensure that reaction paths contain a single potential energy barrier, we slightly displace atoms at the saddle point along the direction of negative curvature (eigenvector, of the Hessian matrix for potential energy, whose eigenvalue is negative) and minimize energy until any force component is lower than $f_{max}^{IS}$. The two resulting minima are then compared with the initial and final ones, and the path is validated if elements of the two pairs of minima match, i.e., their energy difference is less than $\Delta E_{con}$ and the norm of the displacement field separating them is lower than $\Delta r_{con}$. Finally, we filter activated events to remove duplicates. To do that, we start by computing the activation barrier $\Delta E_b$ ($\Delta E$ in main text) and asymmetric potential energy difference $\Delta E_a$ (difference of energy between the final and initial minima), the indexes $p$ and $q$ of the most displaced atoms at saddle point and final minimum, respectively, and their displacement, $\vec{r}_p$ and $\vec{r}_q$, from the initial minimum to the saddle point and to the final minimum, respectively. Then, in crystalline configurations, we consider two reaction paths $i$ and $j$ equal if $|\Delta E_{b,a}^i - \Delta E_{b,a}^j| \leq \Delta E_{dup}$. In amorphous solids, we additionally impose that $\| \vec{r}_{p,q}^i - \vec{r}_{p,q}^j \| \leq \Delta r_{dup}$. These conditions clearly distinguish unique and duplicate activated events in amorphous solids because we use strict force convergence criteria (low values of $f_{max}^{IS}$ and $f_{max}^{\ddagger}$).

**Arrhenius diffusion parameters in crystalline silicon.** We represent in Supplementary Fig. 2, raw measurements of activated events properties—activation energies versus diffusion pre-exponential factors—, obtained in crystalline silicon configurations. For all potentials, many events have activation energies higher than 2 eV; they correspond to reorganizations of the crystal structure far from the defect, so that an artificially high number of such events would be obtained in a large-size configuration. An overrepresentation of such events would bias the evaluation of the compensation effect. To circumvent this issue, we group transition states according to the GS they occur in, and their activation energy, using uniform energy bins of width 0.4 eV, from 0 to 4.8 eV. This upper limit is close to the highest experimental activation energy reported for silicon in Fig. 1a. For each GS, we average properties of events according to the energy bin they lie in, and then average the obtained local properties over all GSs. Finally, we extract the

compensation factor by fitting thereby computed local average values of logarithms of diffusion pre-exponential factors as a function of activation energies with the compensation law. The typical squared distance of the most displaced atom at final minimum, that we use to compute per-event diffusion pre-exponential factors in silicon crystals, is around 1.3 Å in this material.

**The harmonic transition state theory**. Although the harmonic transition state theory (hTST) has been tested against the compensation effect in the past, unexplained contradictory results were reported. Indeed, compensation for diffusion on the surface of metals was both captured[37] and missed[5,38] by hTST computations. Moreover, the compensation effect was reported for bulk diffusion in aluminum[39], silicon carbide[40], and silica[41], while anti-compensation was shown to govern rates of activated events in a non-physical LJ glass[21]. These contradictions are difficult to interpret, especially because numerical compensation factors are never compared with experimental ones in these works. Here, we apply the hTST by computing dynamical matrices $D_{p\alpha,q\beta} = (\partial^2 V/\partial x_{p\alpha}\partial x_{q\beta})/\sqrt{m_p m_q}$ (with $V$ the total potential energy, $x_{p\alpha}$ the cartesian coordinate of atom $p$ along the axis $\alpha$, and $m_p$ its mass) with finite differences. We use the centered-difference formula at order 2, with normalized finite displacements, multiplied by a factor $\delta$ small enough to converge prefactors (see Supplementary Table 2).

**First-principles computations in aluminum**. We combine numerical computations of the literature—obtained using the density functional theory and the harmonic approximation—of pre-exponential factors and activation energies for self- and impurity diffusion in aluminum. The exchange-correlation energy was estimated using the generalized gradient approximation (GGA) for impurities diffusing via the direct interstitial mechanism[42] (H, N, B, O), and using the local density approximation (LDA) for self- and impurity diffusion mediated by vacancies[39] (Al, Mg, Si, Cu). We exclude carbon from the data set because its diffusion path in aluminum contains multiple barriers of comparable amplitude[42] and, in principle, the compensation effect studied here applies only to processes dominated by a single free energy barrier. We also exclude the data for 3$d$ transition metal impurities obtained using the LDA+U approach[43] since reported activation energies are quite different from experimental results. Finally, we note that the LDA method seems to systematically underestimate diffusion pre-exponential factors compared with the GGA method[39]. Given that activation energies of impurities diffusing through vacancies (obtained using LDA) are in average higher than those of interstitials (obtained using GGA), see Fig. 1a, the compensation factor estimated from our data set might be underestimated because of such a systematic difference between these two methods. This would lead to an even closer correspondence with experimental data.

**Analysis of vibrational contributions to the compensation factor**. To wipe out the dispersion of vibrational responses between activated events in amorphous solids, we follow the local averaging procedure used for crystalline silicon, that is we group transition states according to the inherent state they occur in, and their activation energy, using here non-uniform energy bins. Denoting $I_n$ the $n$th energy interval, we then average properties of all events $e$ lying in $I_n$ and occurring in the inherent state $s$. Finally, we compute the macroscopic limit by averaging the obtained properties over all ISs. We use this local averaging procedure to compute local compensation factors as:

$$\gamma_c(\Delta E_n) = \frac{\log \nu_{n+1}^\star - \log \nu_n^\star}{\overline{\Delta E}_{n+1} - \overline{\Delta E}_n},\qquad(1)$$

where $\Delta E_n$ is the center of the $n$th energy interval, $\overline{\Delta E}_n = \langle\langle\Delta E(e,s)\rangle_{e\in I_n}\rangle_s$, and $\log \nu_n^\star = \langle\langle\log\nu^\star(e,s)\rangle_{e\in I_n}\rangle_s$. Using the hTST formula (see main text) and the fact that all events occurring in a same sample share the same initial minimum, we can write

$$\gamma_c(\Delta E_n) \times \left(\overline{\Delta E}_{n+1} - \overline{\Delta E}_n\right) = \sum_i \left(\log\nu^\ddagger\right)_{i,n} - \left(\log\nu^\ddagger\right)_{i,n+1},\qquad(2)$$

with $\left(\log\nu^\ddagger\right)_{i,n} = \langle\langle\log\nu_i^\ddagger(e,s)\rangle_{e\in I_n}\rangle_s$, and $\nu_i^\ddagger$ positive frequencies at transition states arranged in ascending order. Values of the $i$th frequency over TSs lying in a same activation energy bin are distributed closely around the average frequency $\nu_{i,n}^\ddagger = \langle\langle\nu_i^\ddagger(e,s)\rangle_{e\in I_n}\rangle_s$, so that we can finally compute the cumulative local compensation factor $\gamma_c(\nu, \Delta E_n)$ as:

$$\gamma_c(\nu, \Delta E_n) \simeq \log\left(\prod_i \nu_{i,n}^\ddagger/\nu_{i,n+1}^\ddagger\right)/\left(\overline{\Delta E}_{n+1} - \overline{\Delta E}_n\right),\qquad(3)$$

where the product runs over indexes $i$ such that $0 < \nu_{i,n}^\ddagger \le \nu$. Finally, cumulative shifts of non-normalized VDOSs for TSs in the $n$th energy interval, reported in Fig. 2b, are computed at the frequency $\nu$ as

$$\int_{0^+}^{\nu} \langle\langle\rho^\ddagger(\nu',e,s) - \rho^{IS}(\nu',e,s)\rangle_{e\in I_n}\rangle_s \, d\nu'.\qquad(4)$$

**Local average of participation ratios**. At each IS, we compute the vibrational modes of the dynamical matrix, and calculate their participation ratio[44]. The participation ratio $P(\nu_i)$ of the $i$th eigenmode, with frequency $\nu_i$, expresses as:

$$P(\nu_i) = \frac{\left(\sum_p \|\mathbf{e}_{i,p}\|^2\right)^2}{N_{at}\sum_p\|\mathbf{e}_{i,p}\|^4},$$ where sums run over the total number of atoms, $N_{at}$, and $\mathbf{e}_{i,p}$

is the displacement vector of atom $p$ in the $i$th mode. The participation ratio measures the fraction of atoms that significantly contributes to a given eigenmode; it varies between $1/N_{at}$ (when only one atom moves) and 1 (when all atoms contribute equally). Finally, we average results over uniform frequency bins and over all ISs.

## Data availability
The data that support the findings of this study are available from the corresponding authors.

## Code availability
The ARTn code is distributed freely upon request to the corresponding authors.

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

## Acknowledgements

The authors are grateful for extensive computational resources provided by Calcul Quebec/Compute Canada. N.M. acknowledges the support from the director grant from the Trottier Energy Institute, at Polytechnique Montreal, and a Natural Sciences and Engineering Research Council of Canada Discovery Grant.

## Author contributions

S.G. and N.M. conceived the project. S.G. collected experimental data. S.G. and A.C.-R. performed numerical simulations. All authors contributed to data analysis and interpretation. S.G. and N.M. wrote the paper.

## Competing interests

The authors declare no competing interests.
