## [Peer Review File · Nature Communications]

REVIEWERS' COMMENTS

Reviewer #1 (Remarks to the Author):

The compensation effect is an interesting phenomenon that has remained largely unexplained for decades. For thermally activated processes in closely related systems, it is often the case that when the activation energy increases, the pre-exponential factor tends to increase exponentially in a way that compensates. A consequence is that there is one temperature at which all the rates are roughly the same -- i.e., on an Arrhenius plot of the different rates, all the lines with their different slopes and intercepts converge (cross) at one temperature or narrow range of temperatures. It is striking and somewhat mysterious.

The point of this paper is to offer an explanation for this compensation effect. The authors study some systems for which they can create a very large database of reaction pathways, and perform a detailed dissection of the contributions of normal modes in the system to increasing or decreasing the pre-exponential factor as a function of the activation energy. The analysis is somewhat involved, but the physical explanation, e.g., for silicon, is that "On average, directional covalent bonds in amorphous silicon limit local coordination number and, thereby, result in a very open structure where activation typically proceeds through the stretching of bonds, which softens vibrational modes. Since higher barriers are generally associated with more bonds being stretched further, additional softening accompanies them, explaining the origin of the compensation effect in this solid." They discuss other cases as well.

I believe this paper comes closer than I have previously seen in offering a meaningful explanation for the compensation effect. As mentioned above, the analysis is somewhat involved, but the result is a more physics-based understanding, including a rationale for the less common case of the reverse compensation effect. This paper may be appropriate for Nature Communications. Below I list some concerns about the paper, some of which are more important than others.

1) The title does not mention the compensation effect. Am I confused about the point of the paper?

Thank you for the feedback. We had played with a few titles and landed on the one you had initially seen. Following your comment we have revised the title as follows :

« Enthalpy-entropy compensation of atomic diffusion originates from softening of low frequency phonons ».

2) The abstract contains this sentence: "However, no physical model of entropy has ever been successfully tested against experimental data. I find this to be a somewhat dubious claim, although perhaps it depends on how one defines "success". The next sentence in the abstract states "Here, we solve this decades-old problem by demonstrating that atomistically computed harmonic vibrational entropy barriers account for most of compensation effects in silicon and aluminum." What bothers me about this sentence is that it implies that the important advance is showing that computationally evaluated rates (i.e., computed activation energies and pre-exponential factors) agree with the trends seen in experiments. But this has been done many times in the past by many materials computation

groups. And it has been known that both physical (experimental) and model systems can show the compensation effect.

We agree that the definition of « success » can vary and have modified the text and the list of references accordingly. We still believe that it is worth clarifying our statements and our choice of references.

To the best of our knowledge, while there has been experimental studies which gave strong evidence that the compensation effect is a physical law and numerical studies which demonstrated that, for some mechanisms, the harmonic theory held – as is stated by the Referee –, it is the first time that contributions of harmonic vibrational entropy are shown to quantitatively account for experimental compensation effects governing a thermodynamic or kinetic process controlled by a single free energy barrier, in any field of study. Let us recall the steps we follow to do so, and the originality of our results at each step:

Step 1. Determination of the experimental compensation factors, γ_c^{exp} , for the diffusion of impurities in single crystals of silicon and aluminium.

The question of whether the compensation effect bears some fundamental physical significance or merely results from experimental errors remains an active subject of discussion¹⁻⁵. Our study, building on state-of-the-art experimental diffusion data ranging over almost 5eVs in silicon and 3eVs in aluminium, debunks the idea that compensation simply results from experimental errors in these two important materials. In Methods, we added a reference to a recent study of hydrogen diffusion in minerals that meticulously addresses this question and also concludes that compensation is a physical rather than statistical phenomenon⁶.

Step 2. Computation of harmonic compensation factors, γ_c^{harm} , for processes selected at step 1.

In Methods, we give a list of numerical studies that discussed the compensation effect for impurity diffusion in the bulk or at the surface of crystalline solids, within the harmonic approximation. While compensation is observed in some of these studies, it is explicitly concluded in others that the harmonic approximation fails at capturing the compensation effect; anti-compensation is even obtained in one of the listed studies. However, since none of these studies compare numerical compensation factors to experimental ones, or provide an explanation of why compensation should or should not emerge from the harmonic approximation, we believe that it is impossible to assess the rôle of harmonic vibrational entropy on the compensation effect from the literature.

The computation of harmonic compensation factors from simulations is a difficult task. Indeed, diffusion pre-exponential factors may fluctuate wildly around the compensation law, so that meaningful estimations may only be obtained with many mechanisms, extending on large ranges of activation energies. In silicon, we manage to reach the same interval of almost 5eVs than that of experimental results by introducing defects in the lattice structure and probing all reaction pathways with an open-ended saddle point search method (we detail the rationale behind this procedure in the response 7 to reviewer 1). In aluminium, we reach a range of 1.15 eV, with 8 data points, by collecting DFT data from different sources in the literature.

Step 3. Comparison of experimental and numerical results.

If $\gamma_c^{\text{exp}}/\gamma_c^{\text{harm}}$ is larger than 50%, then the comparison is successful, in the sense that most of compensation effects originate from the harmonic entropic contributions evaluated numerically. This definition of success is based on our observation that harmonic entropic contributions systematically underestimate the compensation effect. One could argue that an other contribution, say anharmonicity, counteract the effect of harmonic entropy, and all of compensation effect comes, for instance, from variations of electronic entropy ($\gamma_c^{\text{anharm}} = -\gamma_c^{\text{harm}}$, and $\gamma_c^{\text{exp}} = \gamma_c^{\text{elec}}$). This would be highly speculative and not consistent with existing numerical evaluations of vibrational and electronic contributions to formation entropies and diffusion coefficients^{7,8}.

We find that the harmonic approximation reproduces around 70-80% of compensation effects. It means that 20-30% are missing, and we hope further research will be carried out to understand the origin of these additional entropic contributions.

We insist that we could not find any study offering a similar demonstration in whole of the compensation literature. In this context, we believe our abstract, being unambiguously focused on compensation for impurity diffusion in single crystals, is rather cautious. Of course, if the reviewer believes that we missed an essential reference, we will include it and modify the text accordingly.

To avoid any confusion, we modified the first sentence quoted by the reviewer to: « However, no physical model of entropy has ever been successfully tested against experimental compensation data. »

In the conclusion, we cite papers dealing experimentally or theoretically with the compensation law for processes governed by a single free energy barrier. Our goal is to provide references to compensation phenomena that could likely be captured within numerical simulations, with the sole use of the harmonic approximation, following the three steps described above. We added two references, one for polaron hopping in crystals⁹ and the other for thermally activated motions of molecules at the surface of metals¹⁰.

3) In the abstract as well as in the early part of the paper (page 2), the authors refer to ΔS as an "entropy barrier", analogous to the energy barrier ΔE . However, in the expression for a rate, while ΔE , which is indeed a barrier, appears in the exponential with a negative sign, ΔS appears in the exponential with a positive sign, so it would seem to be better to refer to it as the opposite of a barrier, or to refer to negative ΔS as the entropic barrier. There is no issue that I noticed with equations being wrong there or later in the paper, but this terminology creates some unnecessary confusion.

Thank you, we now use the word « contribution » instead of « barrier » for entropy.

4) Near the bottom of page 2, referring to the rate expression, the authors state "Although resting on thermodynamical principles and being compatible with the Arrhenius law, this expression remains phenomenological in the absence of any analytical expression for the attempt frequency ν and the entropy barrier to migration ΔS_m , which has made sound predictions of Arrhenius pre-exponential factors impracticable so far" Perhaps I am misunderstanding, but this seems like a direct statement that previous workers have been unable to compute pre-exponential factors, or to compute them accurately.

However, many groups have computed pre-exponential factors using the same methodology as in this paper, and often they are shown to be accurate. This is related to point (2) above.

It is true that the classical harmonic approximation has been shown to provide accurate estimations of pre-exponential factors on many occasions; examples may be found in our paper, which relies entirely, for aluminium, on harmonic prefactors found in the literature. Yet, it does not prove that this approximation accurately predicts activation rates in any situation, or accounts for any kinetic or thermodynamic phenomenon. Indeed, in addition to neglecting anharmonic and electronic contributions to entropy, it rests on a set of hypotheses whose quantitative characterization is still an active field of research¹¹. In fact, the accuracy of estimations based solely on harmonic vibrational entropy are often questioned or even shown to be invalid in the literature^{7,12-15}. It has even been argued that anharmonicities of solids' potential energy landscapes along diffusion pathways need to be taken into account to recover the compensation law¹⁶. The question of which mechanistic framework to adopt in order to capture compensation effects was therefore opened before our study. This is so true that, due to the absence of such a framework, current explanations of the compensation effect either draw on phenomenological descriptions of the diffusion process, or even ascribe it to experimental errors.

We modified the sentence quoted by the reviewer: « Although resting on thermodynamical principles and being compatible with the Arrhenius law, this relation remains phenomenological if not complemented by a mechanistic framework within which to evaluate and interpret the different terms it is composed of. Establishing such a framework is particularly difficult for entropic contributions, as they may arise from diverse physical phenomena at the atomic or electronic scales^{7,8,15,16}, without any general rule to assess which dominates. »

5) The y axis of Fig. 1a should be labeled as $\log_{10}(\text{prefactor})$, not a linear scale in the prefactor.

We modified the label, thank you.

6) In Fig. 1a, why are the points not shown (nor discussed in the Methods or supplementary section) for the EDIP or SW potentials. Is it because they showed much greater scatter?

No, it is not the case, but it is true we should have shown these points. We added two sub-figures (one per potential) in Supplementary Fig. S2, with raw data and local averages. In average, and within our finite set of activated events, data best follows the compensation law with the EDIP potential, while comparable deviations are obtained with SW and Tersoff potentials.

7) At the bottom of page 5 (and Fig. 1a), where the results for silicon are compared to experimental results, I am struck by the fact that while the experimental results are for the diffusion rates of many different impurity species, the modeling results are for pure silicon mechanisms, with no impurities.

This approach draws on the experimental observation that a single pair of compensation parameters applies to all impurities in silicon and aluminium, whether diffusion is assisted by defects or proceeds via the interstitial mechanism. The compensation effect may therefore be studied by focusing only on

the activation of pure silicon mechanisms. Note that we also explore numerical data in aluminium, that take into account the nature of impurities as well as the concentration of defects in the computation of Arrhenius diffusion coefficients. We now put more emphasize on this point when introducing our numerical study.

The source of the compensation law, as provided by our work, also explains this unexpected result : since the compensation law is dominated by the change in the phonon spectrum of the host solid, not of the impurity, it is relatively unaffected by the nature of the impurity. A phenomenon that had not been explained before.

This was mentioned in our manuscript but we have clarified it and explained more directly that delocalized vibrational modes, controlled mainly by the properties of the host solid lattice, dominate the compensation effect over localized modes, that may be more affected by interactions between the host solid and impurities.

More precisely, we have modified the paragraph on page 4 that now starts « Our numerical study... » as well as the ending of the paragraph starting with « How does compensation emerge ? », on page 11.

8) A statement similar to the ones in points (2) and (4) is made in the Methods subsection entitled "The harmonic Transition State Theory".

We slightly precised our statement in Methods, according to our responses above : « These contradictions are difficult to interpret, especially because numerical compensation factors are never compared to experimental ones in these works. »

9) page 21: In the definition of the participation ratios, it may not be obvious to the reader (it was not to me) why the coefficients should be taken to the 4th power, and inverted. Perhaps this can be explained a bit and/or referenced.

We added a comment and a reference in Methods.

10) In Supplementary Figure S1, there is an opportunity to show the crossing of all the Arrhenius lines caused by the compensation effect if the fit lines are extended to the left and the x axis ($1/T$) is taken all the way to zero. The lines come together at about 0.25 on the x axis. In my opinion, this could make a compelling figure to include in the main text, although I realize there are space limitations.

Thank you for the suggestion, we have modified Fig. S1 to show the crossing of Arrhenius lines at around $(k_B\gamma_c)^{-1}$ ($\sim 3870K$ in silicon).

Reviewer #2 (Remarks to the Author):

The manuscript 'Entropic acceleration of atomic diffusion by softening of low frequency phonons' by Simon Gelin, Alexandre Champagne-Ruel and Normand Mousseau makes an important contribution to our understanding of diffusion of impurities in solids. It provides a convincing explanation for the 'compensation effect' where the prefactor in the diffusion constant is observed to increase with activation energy.

The manuscript is clearly written and well supported by persuasive computational evidence in good qualitative agreement with experimental data from literature. The breakdown in contributions made by different frequencies to the compensation factor in Fig. 2c is particularly convincing.

In my view the topic is highly suitable for publication in Nature Communications. However, before I can recommend publication I have a few comments the authors should consider:

1. As originally proposed, the EDIP, Stillinger-Weber and Tersoff potentials were parameterised only for pure silicon. The authors appear to have either used modified versions or to have refit the models themselves to include the impurity species considered here. I could not find details of how this was done or of the resulting parameters; this needs to be included in the revised manuscript to enable reproducibility of the research.

We simulate only pure silicon systems, with structural defects but without any chemical impurity, so that we do not need to modify the silicon potentials. Our method consists in searching for saddle points from the ground states of silicon crystals with defects, and then in associating a diffusion coefficient to each of the probed activation path, within the harmonic transition state theory. We argue from the analysis of experimental data that this method enables to recover the compensation effect because this latter does not depend on the chemical nature of the impurity, nor on whether the diffusion is mediated by defects or not (the kinetic and thermodynamic compensation effects are equivalent); the law of compensation is solely determined by the response of the host lattice to a local deformation corresponding to the introduction of a defect or equivalently to the displacement of the solid from its initial ground state to a neighboring transition state.

This question is related to question 7 by reviewer 1. We did not explain clearly our methodology and its rationale, so we have now modified the introduction to our numerical study to improve this point, we have explained why the nature of impurities has no impact on the compensation law based on our analysis of the source of compensation, and we have also added a sentence in Methods to make it clear that we do not include impurity species in our study of silicon.

2. The straight lines in Fig. 1a. appears to indicate the diffusion prefactor as plotted is linearly, rather than exponentially, proportional to the activation energy. I assume the vertical scale should in fact be labelled ' $\log_{10}(\text{diffusion prefactor})$ ' or similar. In the panel showing the DFT data for aluminum, it would be helpful to use a color code to show which data came from LDA calculations and which from GGA ones, to help interpret the scatter in these data.

We corrected the label, thank you. We also added information about approximations of the exchange-correlation energy functional in Fig. 1a.

3. At a first read, the meaning of the colored dots in Fig 1b. is rather opaque I suggest adding a brief explanation to the caption of Fig 1, directing the reader to Fig 2 and/or later in the main text.

Thank you for the suggestion, we added a brief explanation.

4. The value for the maximum force component of $f_{\max}^{\text{IS}} = 3 \times 10^8$ eV given in the Methods section on page 16 appears very large - is this correct, and if so how are the dynamics kept stable during relaxation with such large forces present?

A minus sign is missing, thank you for noticing. We corrected it: $f_{\max}^{\text{IS}} = 3 \times 10^{-8}$ eV.

REVIEWERS' COMMENTS

Reviewer #1 (Remarks to the Author):

This is a follow-up report on this paper. As stated in my original report, this paper presents a notable advance on understanding the striking and somewhat mysterious phenomenon known as the compensation effects, although I pointed out some concerns that I had. I believe the authors have adequately addressed these concerns, and I feel this paper is now appropriate for publication in your journal.

Reviewer #2 (Remarks to the Author):

The authors have responded in detail to my comments and to those of the other reviewer. I am now happy to recommend publication.

REFERENCES

1. Dunstan, D. J. The role of experimental error in arrhenius plots: Self-diffusion in semiconductors. *Solid State Communications* **107**, 159–163 (1998).
2. Fisher, D. J. *The Meyer-Neldel Rule*. (Trans Tech Publications Limited, 2001).
3. Barrie, P. J. The mathematical origins of the kinetic compensation effect: 1. the effect of random experimental errors. *Phys. Chem. Chem. Phys.* **14**, 318–326 (2012).
4. Barrie, P. J. The mathematical origins of the kinetic compensation effect: 2. the effect of systematic errors. *Phys. Chem. Chem. Phys.* **14**, 327–336 (2012).
5. Yelon, A., Sacher, E. & Linert, W. Comment on “The mathematical origins of the kinetic compensation effect” Parts 1 and 2 by P. J. Barrie, *Phys. Chem. Chem. Phys.*, 2012, 14, 318 and 327. *Physical Chemistry Chemical Physics* **14**, 8232 (2012).
6. Jones, A. G. Compensation of the Meyer-Neldel Compensation Law for H diffusion in minerals. *Geochemistry, Geophysics, Geosystems* **15**, 2616–2631 (2014).
7. Satta, A., Willaime, F. & Gironcoli, S. de. Vacancy self-diffusion parameters in tungsten: Finite electron-temperature LDA calculations. *Physical Review B* **57**, 11184–11192 (1998).
8. Metsue, A., Oudriss, A., Bouhattate, J. & Feaugas, X. Contribution of the entropy on the thermodynamic equilibrium of vacancies in nickel. *The Journal of Chemical Physics* **140**, 104705 (2014).
9. Emin, D. Generalized Adiabatic Polaron Hopping: Meyer-Neldel Compensation and Poole-Frenkel Behavior. *Physical Review Letters* **100**, (2008).
10. Gehrig, J. C. *et al.* Surface single-molecule dynamics controlled by entropy at low temperatures. *Nature Communications* **8**, (2017).
11. Gesù, G. D., Lelièvre, T., Peutrec, D. L. & Nectoux, B. Jump Markov models and transition state theory: the quasi-stationary distribution approach. *Faraday Discussions* **195**, 469–495 (2016).
12. Pandey, K. C. & Kaxiras, E. Entropy calculation beyond the harmonic approximation: Application to diffusion by concerted exchange in Si. *Physical Review Letters* **66**, 915–918 (1991).

13. Ryu, S., Kang, K. & Cai, W. Entropic effect on the rate of dislocation nucleation. *Proceedings of the National Academy of Sciences* **108**, 5174–5178 (2011).
14. Nguyen, L. D., Baker, K. L. & Warner, D. H. Atomistic predictions of dislocation nucleation with transition state theory. *Physical Review B* **84**, (2011).
15. Swinburne, T. D. & Marinica, M.-C. Unsupervised Calculation of Free Energy Barriers in Large Crystalline Systems. *Physical Review Letters* **120**, (2018).
16. Marinica, M.-C., Barreteau, C., Spanjaard, D. & Desjonquères, M.-C. Diffusion rates of Cu adatoms on Cu(111) in the presence of an adisland nucleated at fcc or hcp sites. *Physical Review B* **72**, (2005).